# Prevalence and its associated factors of medical error reporting among healthcare professionals in Ethiopia: Systematic review and meta-analysis

Yeshiambaw Eshetie[1]*, Tigabu Munye Aytenew[1], Yirgalem Abere[1], Bekalu Mekonen Belay[1], Tekalign Amera[1], Mengistu Ewunetu[1], Gebrie Kassaw Yirga[1], Yohannes Tesfahun[2], Melese Kebede[2], Demewoz Kefale[3]

1 Department of Adult Health Nursing, College of Health science, Debre Tabor University, Debre Tabor, Ethiopia, 2 Department of Emergency and Critical Care Nursing, College of Health Sciences, Debre Tabor University, Debre Tabor, Ethiopia, 3 Departmenyt of Pediatrics and Child health Nursing, College of Health Sciences, Debre Tabor University, Debre Tabor, Ethiopia

* yeshiambaweshetie@gmail.com

## Abstract

### Introduction

Medical error refers to a mistake made by a healthcare professional that poses a significant threat to patient safety worldwide. Reporting these errors is crucial for reducing healthcare-related mistakes. Despite several studies on medical error reporting and its associated factors among health professionals in Ethiopia, the national prevalence and contributing factors are not well established.

### Methods.

We conducted a systematic review and meta-analysis of cross-sectional studies assessing the prevalence of medical error reporting and its associated factors among healthcare professionals in Ethiopia. An extensive literature search was performed from April 10 to June 10, 2024, using databases such as Google Scholar, Web of Science, and PubMed, along with a manual search. The pooled prevalence was calculated using a random-effects model.

### Results

Out of 1233 studies retrieved from databases, only 24 studies involving a total of 6,745 healthcare professionals were included in the analysis. The overall pooled prevalence of medical error reporting was 42.66% (95% CI: 33.19, 52.13; $I^2 = 98.79\%$, $p < 0.01$). Factors significantly associated with medical error reporting included being trained (AOR = 3.25, 95% CI: 1.79, 4.70), fear of administrative sanctions (AOR = 0.40, 95% CI: 0.04, 0.76), lack of feedback (AOR = 0.86, 95% CI: 0.16, 1.55; $p < 0.02$), increased work experience (AOR = 2.90, 95%CI: 1.25, 4.54),

**Data availability statement:** All relevant data are within the manuscript and its supporting information files

**Funding:** The author(s) received no specific funding for this work.;

**Competing interests:** The authors have declared that no competing interests exist

**Abbreviations:** M.E, Medical Error; CI, Confidence Interval; AOR, Adjusted Odds Ratio; JBI, Joanna Briggs Institute; PRISMA, Preferred Reporting Item for Systematic review and Meta-analysis

female professionals (AOR = 2.22, 95% CI: 0.10, 4.34), and higher education status (AOR = 3.20, 95%CI: 1.10, 5.30).

## Conclusion

Medical error reporting among healthcare professionals in Ethiopia is relatively low, primarily due to inadequate training, fear of consequences, and lack of feedback. Targeted interventions such as training programs and the creation of a non-punitive error reporting environment are needed to improve reporting practices.

## Introduction

Medical errors (MEs) refer to mistakes made by healthcare professionals during diagnosis, treatment, care, or monitoring [1,2], often resulting from failures to execute planned actions or the use of inappropriate plans to achieve desired outcomes [3]. They encompass a diverse group of errors, such as medication errors, adverse reactions, and/or any incidents affecting patients [4–6]. These errors occur regularly and pose significant threats to patient safety globally [3]. Although the majority of errors do not lead to apparent adverse effects, a considerable number of patients suffer from permanent harm or die each year worldwide due to these mistakes [6,7]. Medical errors are a leading cause of health issues, with 134 million mistakes reported each year in developing countries, causing 2.6 million deaths [8,9].

Establishing a patient safety culture in healthcare can be achieved by implementing strategies that effectively reduce errors and their recurrence. One key strategy is medical error reporting, which plays a crucial role in reducing healthcare professional-related mistakes [10,11]. Medical error reporting involves documenting and disclosing errors to the relevant body or colleagues. The data collected through reporting can then be used to investigate the types of medical errors, enhancing our understanding of their causes and analyzing the root factors. Moreover, error reporting would enable professionals to learn from their errors in the same healthcare setting while also contributing to a valuable record that serves as a learning resource worldwide [12,13]. Therefore, it is important to capture, track, and analyze medical errors at every healthcare institution.

Medical error reporting includes medication error, adverse drug events, and incident reporting [14,15]. Medication error reporting reveals specific types of mistakes, including errors related to prescribing, dispensing, or administering medications [16,17], whereas incident reporting encompasses a broader range of errors that could lead to patient harm, including but not limited to medication mistakes, equipment failures, or patient falls [18,19]. Similarly, adverse drug event reporting involves documenting any harmful event related to the use of medications, encompassing both mistakes and adverse reactions [20,21]. All three types of error reporting contribute to enhancing patient safety and ensuring quality care within the institution [2,22]. Implementing a comprehensive error reporting system in hospitals helps collect diverse data, improving the identification of patterns, trends, and root causes in medical

errors [23]. This holistic approach enhances the understanding of how different type of errors are related to each other and can reveal underlying systemic issues that might not be apparent when analyzing each category separately. Moreover, this comprehensive medical error reporting fosters collaboration among physicians, nurses, pharmacists, and other healthcare professionals to identify deficiencies causing errors and implement preventive strategies [2].

Despite the significant benefits of error reporting, underreporting remains alarmingly prevalent worldwide. For instance, the rate of underreporting in Iran ranges from 50.5% to 95.5%, with participants indicating that they experienced at least one medical error within a year [24,25]; in Uganda; it is 64.6% [26]; in Saudi Arabia, 65.7% [27]; and in United Kingdom, 30.7% [28]. Primary studies have also been conducted across various regions of Ethiopia: in Oromia, the error reporting rate is 29.2% [29]; in Addis Ababa, it is 27.4% [30]; and in Tigray, it is 32.1% [31].

Studies have indicated that several factors are associated with the underreporting of medical errors [32,33]. Evidence in Saudi Arabia and Korea, for instance, found approximately ninety-six barriers to error reporting, which were classified into individual and organizational factors. Among the most frequently reported barriers were a poorly designed reporting system and inadequate patient safety leadership [34,35]. Moreover, a systematic review conducted in the USA identified several barriers to medical error reporting, which were categorized into themes. These themes included fear of consequences, lack of feedback, poor understanding of the importance of error reporting, workload, lack of reporting system, and personal factors [36]. In Ethiopia, barriers have also been identified, including lack of feedback, a non-punitive environment, lack of training, fear of administrative sanctions, absence of a reporting system, and lack of communication openness [37,38]. Although numerous studies have assessed the prevalence of medical error reporting and its associated factors among health professionals in various regions of Ethiopia, their findings have been inconsistent, and none have reported the national prevalence of error reporting practices. Therefore, this study aimed to determine the overall pooled prevalence of medical error reporting in Ethiopia and its associated factors. The findings will provide evidence to help healthcare policymakers develop policies that encourage error reporting, guide clinicians in improving patient safety, and support healthcare managers in implementing systemic improvements to ensure quality care within healthcare institutions.

## Methods and materials

### Reporting and registration protocol

The results of this review were reported according to the Preferred Reporting Items for Systematic Reviews and Meta-Analyses (PRISMA) [39] guidelines (S1 File). The protocol for the systematic review registered with the International Prospective Register of Systematic Reviews (PROSPERO) under registration number CRD42024545717. This registration helps prevent duplicate reviews by different researchers, reduces reporting bias, and enhance transparency.

### Eligibility criteria

This study included all observational studies conducted among healthcare professionals in Ethiopia that reported the prevalence of medical error reporting and/or at least one factor associated with error reporting and published in English. However, articles lacking abstracts or full texts, as well as systematic reviews, meta-analyses, and qualitative studies, were excluded.

### Databases and search strategy

Articles were retrieved from Google Scholar, Web of Science, and PubMed using specific search terms and phrases. Additionally, a manual search was conducted to identify gray literature. The Condition, Context, and Population (CoCoPoP) search design was employed. The relevant search terms comprised a combination of Medical Subject Headings (MeSH) and free-text keywords, using Ethiopia as a search filter. Boolean operators (AND, OR) were used to create the search string (S2 File). The searches were conducted without language restriction from April 10 to June 10, 2024. For instance,

the search string for one database was "Medical errors" AND "Reporting" OR "disclosure" AND "magnitude" AND "associated factors" OR "predictors" OR "determinants" AND "healthcare professionals" OR "nurses" OR "doctors" OR "pharmacists" AND "Ethiopia"

## Study selection procedures

All retrieved studies were imported into EndNote Reference Manager (version 7) to remove duplicates. Two authors (B.M. and D.K.) independently screened the titles and abstracts of all articles, followed by a full-text review to assess eligibility. Any disagreements were resolved through discussion with a third reviewer (M.K.).

## Data extraction and quality evaluation

The two independent reviewers (Y.E. and M.K.) extracted the data using a structured Microsoft Excel worksheet. To ensure consistency and accuracy, they performed cross-checks throughout the process. When inconsistencies were observed in the extracted data, the phase was repeated. If there was any disagreement, a third author (D.K.) was consulted to facilitate discussion and help reach a consensus. The first author's name, year of publication, study year, study design, sample size, response rate, prevalence (practice) of error report, effect size, types of error reporting, and area of study were extracted in each article. If any relevant data were unavailable during extraction from the articles, the assessors contacted the corresponding author for clarity. The article was excluded if adequate responses were not obtained based on exclusion criteria or handled through the complete case analysis method. But there was no evidence to perform single or multiple imputation since the missing value is less than 5%. The standard error of prevalence for each study was calculated prior to data analysis. After calculating the natural logarithm of the lower and upper limits of the confidence interval (CI) for the adjusted odds ratio, the standard error for each article's adjusted odds ratio was derived before data analysis.

Each study was evaluated for quality using the Joanna Briggs Institute (JBI) Critical Appraisal Checklist for Studies Reporting Prevalence Data [40]. Two independent investigators (Y.E. and M.K.) appraised the quality of included articles. The evaluation of the included studies focused on the appropriateness of the study participants, settings, design, and measurements. Articles were considered to be of low risk or good quality if they scored 50% or higher on the quality appraisal indicators and were included in the review. The quality scores of the evaluated articles ranged from 77.8% to 88.9% (**Table 1**). Therefore, all the included studies were considered low risk or good quality.

## Outcome measures

The primary outcome of this review was the pooled prevalence of medical error reporting among health professionals in Ethiopia. The associated factors of error reporting were also the secondary focus of the review.

## Data analysis

The extracted data were exported to STATA version 17 for statistical analysis. Heterogeneity among the included primary studies was likely reduced by using a random-effects model [41]. A weighted inverse-variance random-effects model [42] was employed to calculate the overall pooled prevalence of medical error reporting and to identify the effect of predictors on medical error reporting. Publication bias was checked using a funnel plot and Egger's test, with a p-value <0.05% indicating the presence of significant publication bias. Heterogeneity was assessed employing $I^2$ statistical test, Galbraith plot, and Cochran's Q statistic. The $I^2$ statistics quantify the percentage of total variation among the included articles. $I^2$ statistics values of 0%, 25%, 50%, and 75% correspond to no, low, moderate, and high heterogeneity, respectively. A p-value of less than 0.05 for the $\chi^2$ test for Cochran's Q statistic was used to indicate significant heterogeneity. Sensitivity analysis was performed to identify the influence of a single study on the overall results of the meta-analysis. A forest plot was used

**Table 1. Quality assessment of the included articles for this study using JBI'S critical appraisal checklist for cross-sectional study, Ethiopia, 2024.**

| Author (Year) | JBI'S Critical Appraisal questions | | | | | | | | | Quality assessment | | Inclusion |
|---|---|---|---|---|---|---|---|---|---|---|---|---|
| | Q1 | Q2 | Q3 | Q4 | Q5 | Q6 | Q7 | Q8 | Q9 | Score (%) | Risk of bias | Included |
| Engeda et al(2016) | Y | Y | Y | Y | N | Y | Y | Y | Y | 8(88.9) | Low risk | ∏ |
| Agegnehu et al(2017) | Y | Y | Y | Y | N | Y | Y | Y | Y | 8(88.9) | Low risk | ∏ |
| Eshete et al(2021) | Y | Y | Y | Y | N | Y | Y | Y | Y | 8(88.9) | Low risk | ∏ |
| Yalew et al(2021) | Y | Y | N | Y | Y | N | Y | Y | Y | 7(77.8) | Low risk | ∏ |
| Kefale et al(2017) | N | Y | Y | Y | N | Y | Y | Y | Y | 7(77.8) | Low risk | ∏ |
| Gidey et al(2020) | Y | Y | Y | Y | Y | Y | Y | Y | N | 8(88.9) | Low risk | ∏ |
| Kassa et al(2019) | Y | Y | Y | Y | N | Y | Y | N | Y | 7(77.8) | Low risk | ∏ |
| Gurmesa et al(2016) | Y | Y | Y | Y | N | Y | Y | Y | Y | 8(88.9) | Low risk | ∏ |
| Shanko et al(2018) | Y | Y | Y | Y | N | Y | Y | Y | Y | 8(88.9) | Low risk | ∏ |
| Hailu et al(2014) | Y | Y | Y | Y | N | Y | Y | Y | Y | 8(88.9) | Low risk | ∏ |
| Zimamu et al(2021) | Y | Y | N | Y | Y | Y | Y | Y | Y | 8(88.9) | Low risk | ∏ |
| Kassa et al(2016) | Y | Y | Y | Y | N | Y | Y | Y | Y | 8(88.9) | Low risk | ∏ |
| Nadew et al(2020) | Y | Y | Y | Y | N | Y | Y | Y | Y | 8(88.9) | Low risk | ∏ |
| Bule et al(2016) | Y | Y | Y | Y | N | Y | Y | Y | Y | 8(88.9) | Low risk | ∏ |
| Asefa et al(2021) | Y | Y | N | Y | Y | Y | Y | Y | Y | 8(88.9) | Low risk | ∏ |
| Jember et al(2018) | N | Y | N | Y | Y | Y | Y | Y | Y | 7(77.8) | Low risk | ∏ |
| Bifftu et al(2016) | Y | Y | Y | Y | Y | Y | Y | Y | N | 8(88.9) | Low risk | ∏ |
| Jifar et al(2022) | Y | Y | Y | Y | N | Y | Y | Y | Y | 8(88.9) | Low risk | ∏ |
| Siraj et al(2022) | Y | Y | Y | Y | N | Y | Y | Y | Y | 8(88.9) | Low risk | ∏ |
| Necho et al(2014) | Y | Y | N | Y | Y | Y | Y | Y | Y | 8(88.9) | Low risk | ∏ |
| Seid MA et al(2018) | Y | Y | Y | Y | N | Y | Y | Y | Y | 8(88.9) | Low risk | ∏ |
| Abay et al(2008) | Y | Y | Y | Y | N | Y | Y | Y | Y | 8(88.9) | Low risk | ∏ |
| Shemsu et al (2024) | Y | Y | Y | Y | N | Y | Y | Y | Y | 8(88.9) | Low risk | ∏ |
| Mulisa et al (2015) | N | Y | Y | Y | N | Y | Y | Y | Y | 7(77.8) | Low risk | ∏ |

Abbreviations **and detail of each questions**: Y = Yes, N = No, Q = Question. The total quality result is calculated by counting the number of Y in each row. Q1: Was the sample frame appropriate to address the target population? Q2: Were study participants sampled in an appropriate way?/Are the patients at a similar point in the course of their condition/illness? Q3: Was the sample size adequate? Q4: Were the study subjects and the setting described in detail?/Are confounding factors identified and strategies to deal with them stated? Q5: Was the data analysis conducted with sufficient coverage of the identified sample?/Are outcomes assessed using objective criteria? Q6: Were valid methods used for the identification of the condition? Q7: Was the condition measured in a standard, reliable way for all participants? Q8: Was there appropriate statistical analysis?/Were outcomes measured in a reliable way? q9: Was the response rate adequate, and if not, was the low response rate managed appropriately

to estimate the effect of independent factors on the outcome variable, with the measure of association reported at a 95% confidence interval.

## Results

### Study selections

An extensive search was conducted, and a total of 1,233 studies were retrieved from databases such as Google Scholar (500), Web of Science (200), PubMed (510), and manual search (13) and from the university research repository online library (10). After removing duplicates (100) and irrelevant studies based on their abstracts and titles (1000), a total of 133 studies were selected for full-text review (S3 File). During the full-text review, 89 studies with inaccessible full-texts were removed. Of the remaining 44 studies, 20 were removed due to differences in design and language. Finally, 24 studies

remained for data extraction to determine the pooled prevalence and associated factors of medical error reporting in Ethiopia [43] (Fig 1).

## Characteristics of included studies

A total of 24 studies were included in this review: 5 of them were about incident reporting [4,37,44–46], 3 of them about medication error reporting [47–49] and the remaining 16 of them were about adverse drug reaction reporting [29–31,38,50–61]. All included studies reported the prevalence of medical error reporting. Regarding study region: twelve of the included studies [4,44,45,48–50,52–55,58,59] were from Amhara; four articles [29,38,51,60] were from Oromia; four articles [30,46,47,61] were from Addis Ababa; two studies [37,57] were from Southern Nations, Nationalities and People's region; one article [56] was from Harare region; and the remaining one study [31] was from Tigray region. In terms of study design all of the included articles were cross-sectional. Institutionally, twenty-one articles were hospital-based studies; two studies were based in health centers; and the remaining one was community-based study. In related to study population for the included articles, almost all articles (eighteen) were on healthcare professionals, fives studies were on nurses and one was on physicians. From the perspective of sample size, the involved articles ranged from 62 [54] to 708 [59], with a total of 6,745 respondents (Table 2).

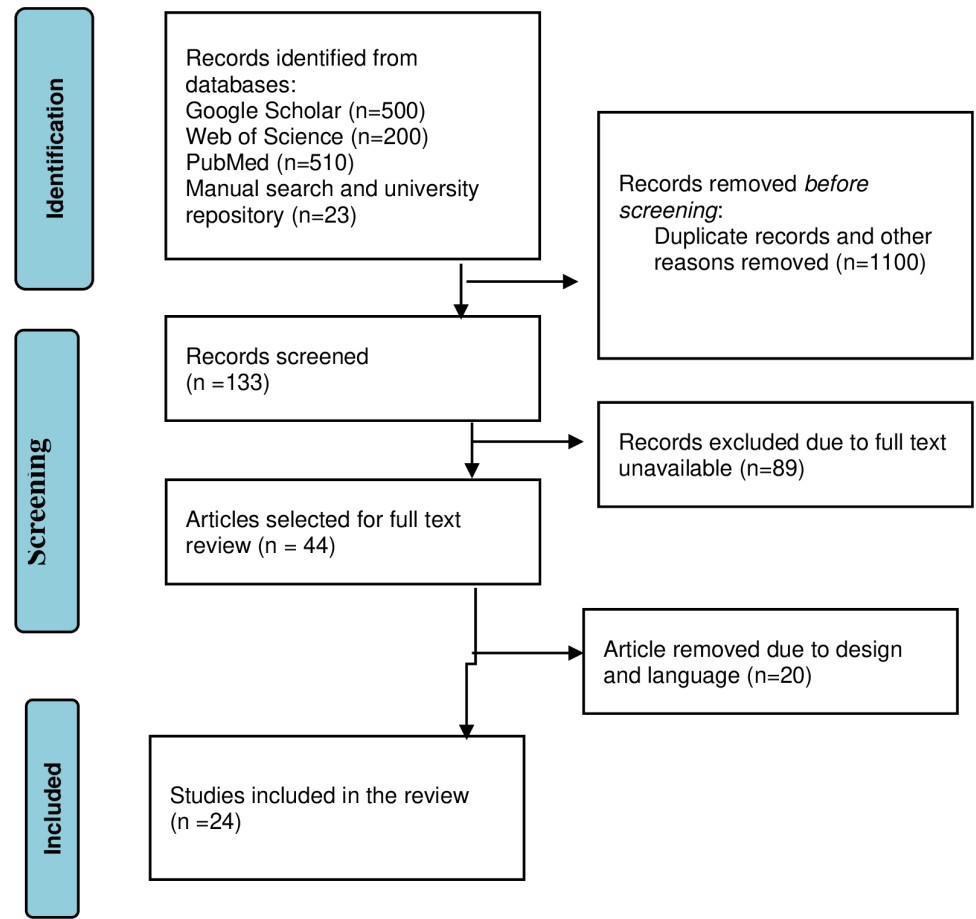

**Fig 1. PRISMA flow chart revealing the study selection process, 2024.**

Table 2. General characteristics of the included primary articles for this study, Ethiopia, 2024.

| NO | Author(Year) | Study Region | Study design | Prevalence | Sample size | Study setting | Study Population | Report type |
|----|--------------|--------------|--------------|------------|-------------|---------------|------------------|-------------|
| 1 | Engeda et al(2016) | Amhara | Cross-sectional | 25.4 | 423 | Hospital | Nurses | Incident |
| 2 | Agegnehu et al(2017) | Addis | Cross-sectional | 30.4 | 697 | Hospital | Health professional | Incident |
| 3 | Eshete et al(2021) | Amhara | Cross-sectional | 31.9 | 319 | Hospital | Nurses | Incident |
| 4 | Yalew et al(2021) | Amhara | Cross-sectional | 12.5 | 319 | Hospital | Health professional | Incident |
| 5 | Kefale et al(2017) | Addis | Cross-sectional | 90.2 | 280 | Hospital | Health professional | Adverse drug |
| 6 | Gidey et al(2020) | Tigray | Cross-sectional | 32.1 | 362 | Hospital | Health professional | Adverse drug |
| 7 | Kassa et al(2019) | Amhara | Cross-sectional | 50 | 120 | Hospital | Health professional | Adverse drug |
| 8 | Gurmesa et al(2016) | Oromia | Cross-sectional | 38.8 | 133 | Health-centre | Health professional | Adverse drug |
| 9 | Shanko et al(2018) | Harar | Cross-sectional | 60.6 | 325 | Hospital | Health professional | Adverse drug |
| 10 | Hailu et al(2014) | Amhara | Cross-sectional | 28.6 | 156 | Hospital | Health professional | Adverse drug |
| 11 | Zimamu et al(2021) | Amhara | Cross-sectional | 74.8 | 215 | Community | Health professional | Adverse drug |
| 12 | Kassa et al(2016) | Amhara | Cross-sectional | 83.3 | 62 | Hospital | Health professional | Adverse drug |
| 13 | Nadew et al(2020) | Addis | Cross-sectional | 27.4 | 422 | Hospital | Physicians | Adverse drug |
| 14 | Bule et al(2016) | Oromia | Cross-sectional | 29.2 | 130 | Hospital | Health professional | Adverse drug |
| 15 | Asefa et al(2021) | Amhara | Cross-sectional | 37.9 | 230 | Hospital | Nurses | Medication |
| 16 | Jember et al(2018) | Addis | Cross-sectional | 57.4 | 423 | Hospital | Nurses | Medication |
| 17 | Bifftu et al(2016) | Amhara | Cross-sectional | 29.1 | 292 | Hospital | Nurses | Medication |
| 18 | Jifar et al(2022) | Oromia | Cross-sectional | 46.7 | 101 | Hospital | Health professional | Adverse drug |
| 19 | Siraj et al(2022) | South Ethiopia | Cross-sectional | 66.2 | 190 | Hospital | Health professional | Adverse drug |
| 20 | Necho et al(2014) | Amhara | Cross-sectional | 16.2 | 708 | Hospital | Health professional | Adverse drug |
| 21 | Seid MA et al(2018) | Amhara | Cross-sectional | 49.1 | 102 | Health-centre | Health professional | Adverse drug |
| 22 | Abay et al(2008) | Amhara | Cross-sectional | 48.2 | 232 | Hospital | Health professional | Adverse drug |
| 23 | Shemsu et al (2024) | South Ethiopia | Cross-sectional | 28.7 | 354 | Hospital | Health professional | Incident |
| 24 | Mulisa et al (2015) | Oromia | Cross-sectional | 30.8 | 150 | Hospital | Health professional | Adverse drug |

## Meta-analysis

### The pooled prevalence of medical error reporting

A total of 24 eligible primary studies [4,29–31,37,38, 44–61] were included in the final meta-analysis, and the pooled prevalence of medical error reporting among health professional in Ethiopia was 42.66% (95% CI: 33.19, 52.13; $I^2 = 98.79\%$, $p < 0.01$) (Fig 2).

### Heterogeneity investigation

The Galbraith plot (Fig 3) and the percentage of $I^2$ statistics from the forest plot revealed substantial heterogeneity among the included primary studies ($I^2 = 98.79\%$, $p < 0.01$). Therefore, subgroup and sensitivity analysis were performed to identify the source of heterogeneity.

### Publication bias

The p-value of the eggers regression test ($p < 0.01$) and the asymmetric distribution of the included primary studies on the funnel plot (Fig 4a) suggest the presence of publication bias. Therefore, a trim-and-fill analysis was performed to address this publication bias (Fig 4b).

**Sensitivity analysis.** Sensitivity analysis was conducted to determine the influence of a single primary study on the overall meta-analysis. As the forest plot revealed that a single primary study estimate is closer to the combined estimate,

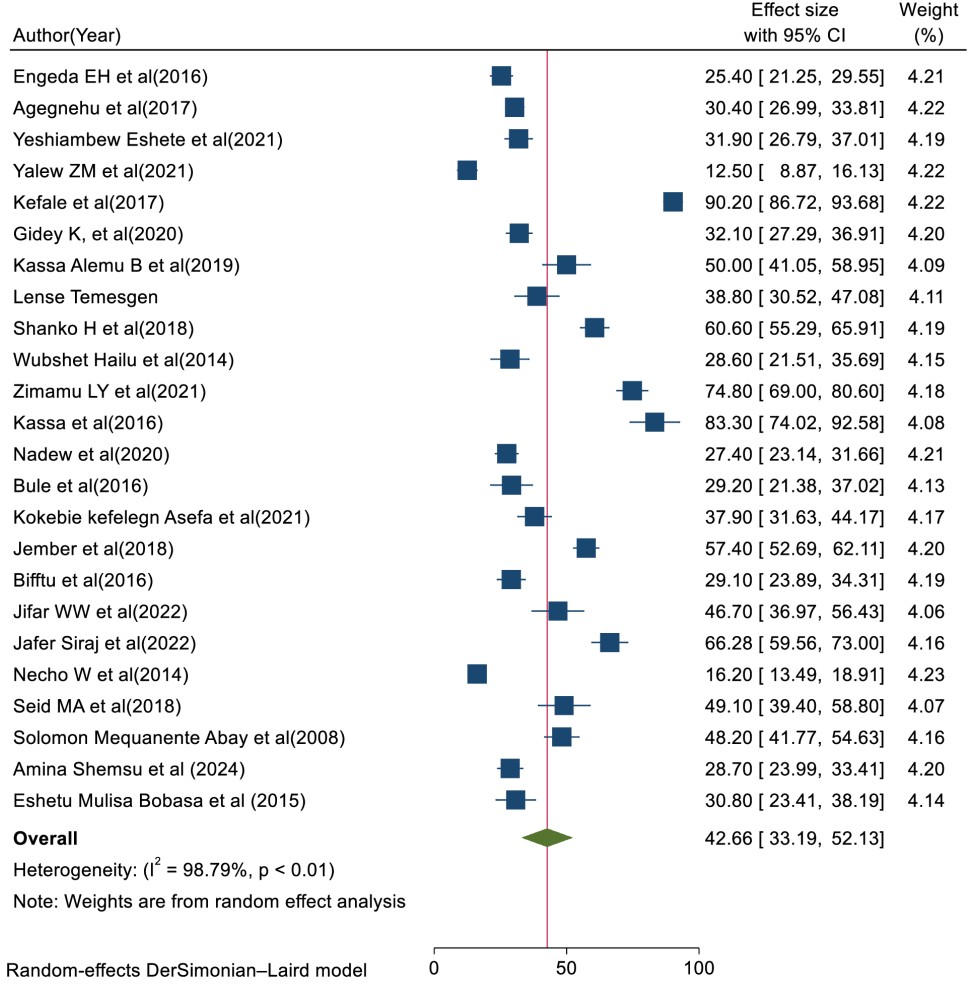

**Fig 2. Forest plot showing the pooled prevalence of medical error reporting with 95%CIs.**

which implied there is no influence of a single primary study on the overall pooled estimate. Therefore, the overall outcome of the meta-analysis is out of the impact of a single primary study as it has been revealed by the forest plot (**Fig 5**).

**Subgroup analysis.** Subgroup analysis was conducted based on the type of medical error reporting (medication error reports, incident reports, and adverse drug reaction reports) and areas of studies (regions in Ethiopia). Therefore, based on the study areas, the highest pooled prevalence of medical error reporting was found in studies conducted in Addis Ababa (51.35, 95% CI: 20.51, 82.20; $I^2 = 99.60\%$, p < 0.01), followed by studies conducted in Southern Ethiopia (47.41; 95% CI: 10.58, 84.24; $I^2 = 98.76\%$; p < 0.01). The lowest pooled prevalence of medical error report was found in studies conducted in the Amhara and Oromia regions of Ethiopia (40.33; 95% CI: 29.04, 51.62; $I^2 = 98.22\%$; p < 0.01) and (35.94; 95% CI: 28.52, 43.36; $I^2 = 69.13\%$, p < 0.01), respectively (**Fig 6**). Based on the above results, we can say that the main source of heterogeneity stems from studies conducted in Addis Ababa, Southern Ethiopia, and Amhara regions. Similarly, based on the type of medical error reporting, the highest pooled prevalence of medical error report was observed in studies focused on adverse drug reaction reporting type (48.25; 95%CI: 34.36, 62.14; $I^2 = 98.98\%$; p < 0.01), followed by studies on medication error reporting (41.51; 95%CI: 23.77, 59.25; $I^2 = 96.96\%$; p < 0.01). The lowest pooled prevalence of medical error reporting was found in studies focused on incident reporting (25.71; 95%CI: 18.32, 33.10;

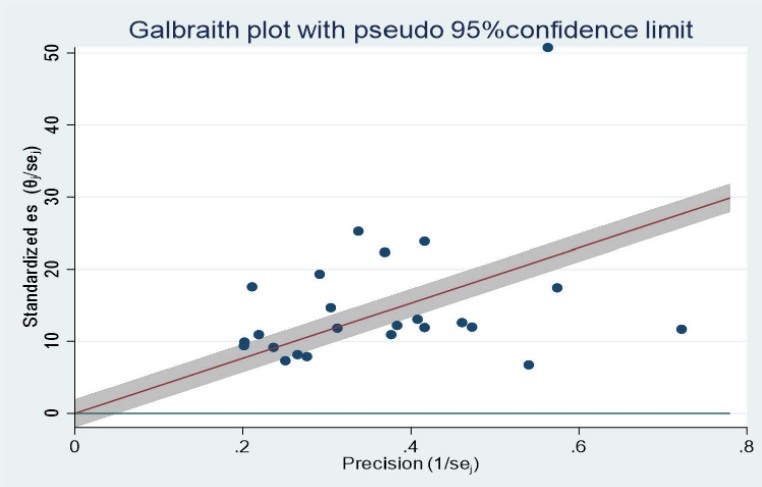

**Fig 3. Galbraith plot revealing the presence of heterogeneity among included articles.**

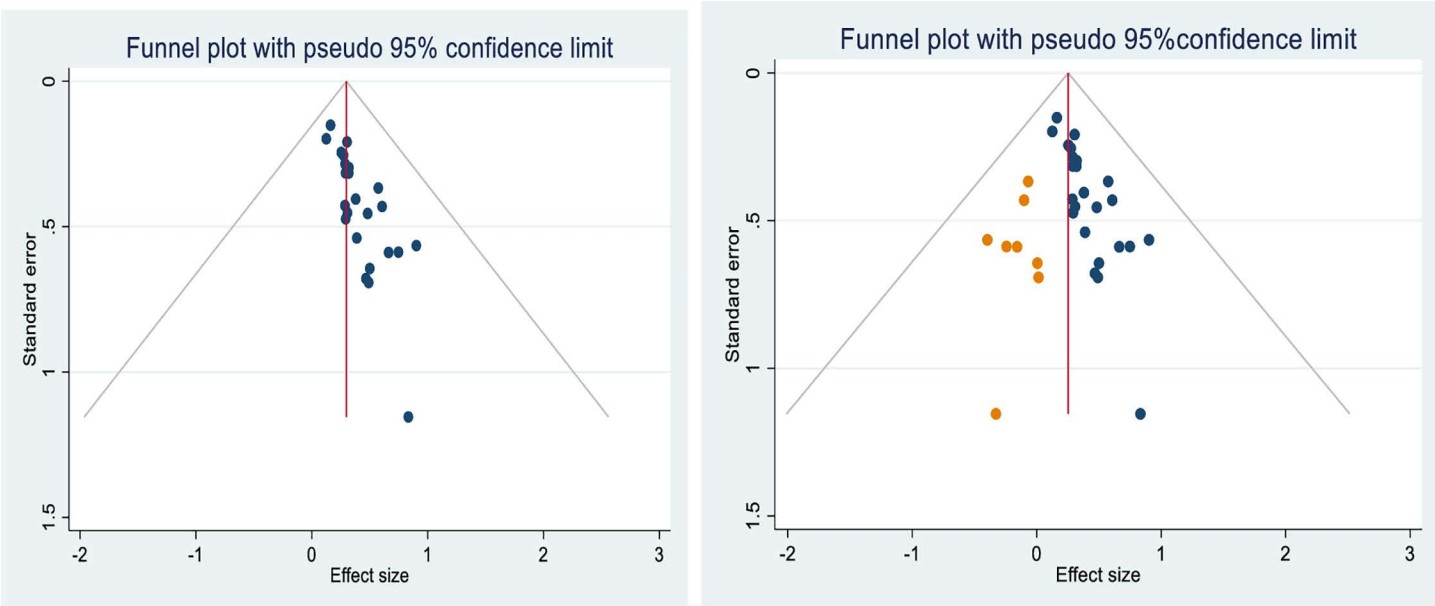

**Fig 4. Funnel plot before adjustment (4a) and after adjustment (4b) using trim-and-fill analysis for publication bias of medical error reporting among health professional, Ethiopia.**

I=93.80%; p<0.01) (**Fig 7**). Therefore, based on the subgroup analysis, the heterogeneity in this study may be attributed to differences in the study area of the primary articles included in this review, with no significant differences found in the type of medical error reporting. Contextual factors, such as errors disclosed in hospitals versus those reported from health and community service centers, may contribute to variations in error reporting practices. On top of that, differences in culture, regulation, and resources across regions can lead to variability in error reporting practice. The variability in this study could also be attributed to differences in the utilization of various error reporting systems (voluntary vs. mandatory

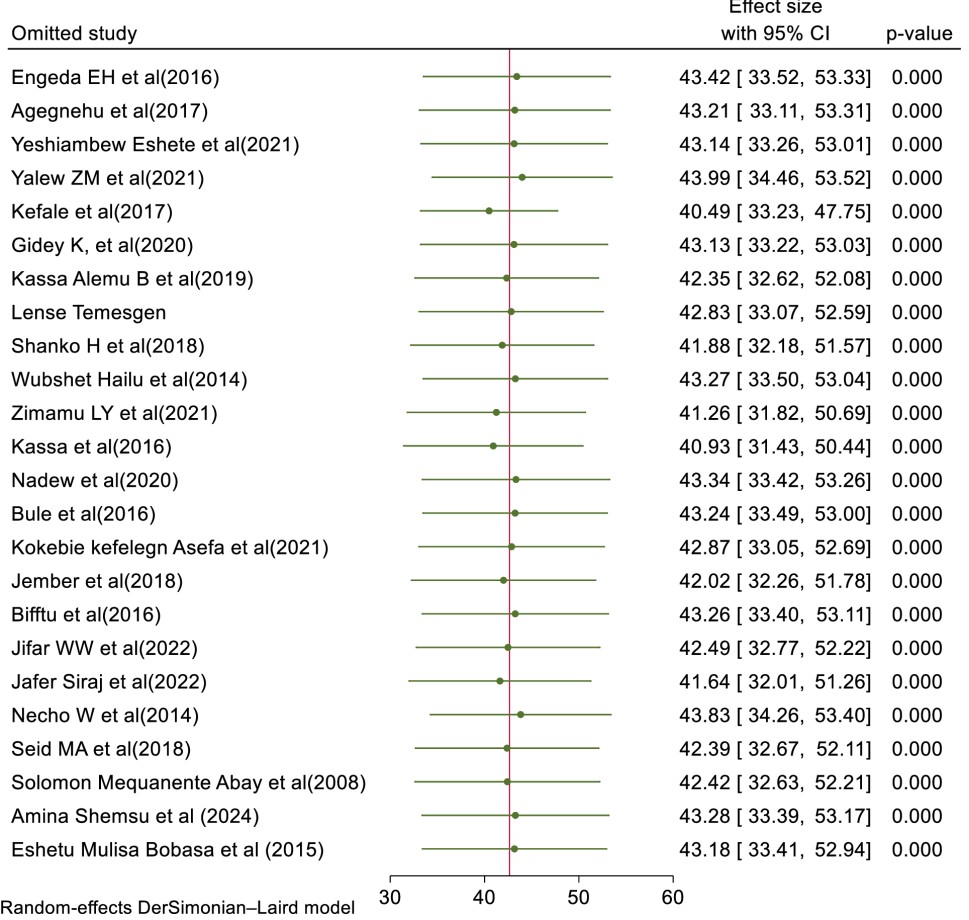

**Fig 5. Sensitivity analysis of the included primary articles of for this study.**

reporting) as well as variations in how errors are defined, measured, and assessed across different healthcare institutions such as hospitals, health centers, and community settings. Additionally, the presence of unmeasured factors that vary across studies may contribute to this heterogeneity.

## Factors associated with medical error reporting

In this review, the pooled determinants of medical error reporting were analyzed. Adjusted factors that were explored in a minimum of two studies need to be compared among the involved articles. Hence, training, fear of administrative sanction, lack of feedback, work experience, professionals' gender, and educational status met the criteria for inclusion in this systematic review and meta-analysis. To assess the relationship between medical error reporting and training, eight articles were involved. Thus, the combination of eight studies reported that training was significantly associated with health professionals medical error reporting (AOR = 3.25, 95% CI: 1.79, 4.70; I² = 95.75%, P < 0.01) (**Fig 8**). This suggests that health professionals who received training were over three times more likely to report errors than those who did not get training. The finding highlights the need for comprehensive training in fostering error reporting. The other four studies also showed that fear of administrative sanction was significantly associated with medical error reporting (AOR = 0.40, 95% CI: 0.04, 0.76; I² = 0.00%; P < 0.03) (**Fig 9**). This indicates that healthcare professionals who fear administrative sanction were

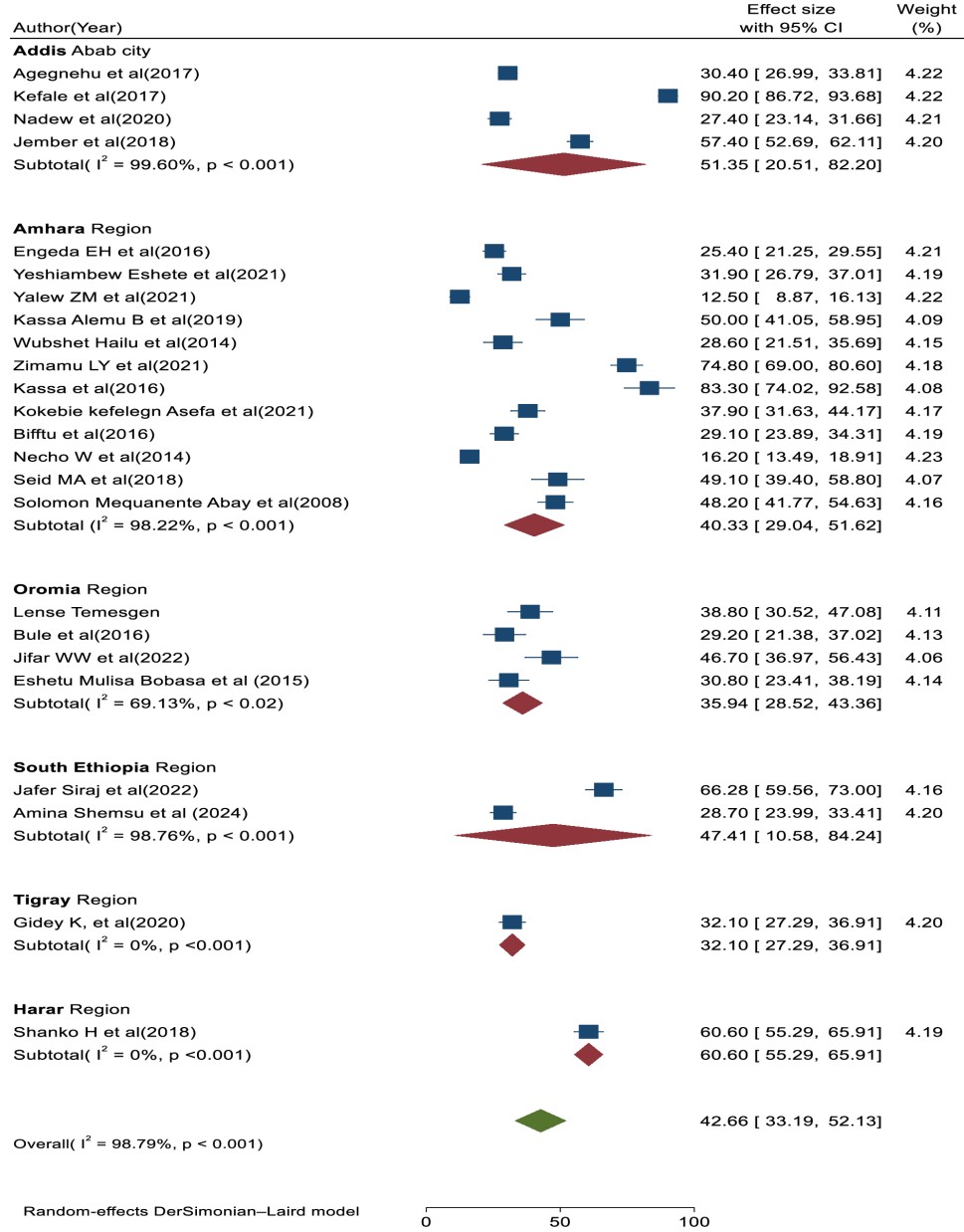

**Fig 6. Forest plot of prevalence of medical error reporting with 95%CIs of the subgroup analysis based on study region, Ethiopia, 2024.**

less likely to report medical error than those who did not fear administrative sanction. The finding implies the need for protection from administrative sanction in promoting error reporting. Similarly, four studies in the included articles reported that the lack of feedback was significantly associated with health professionals medical error reporting (AOR = 0.86, 95% CI: 0.16, 1.55; $I^2$ = 81.30%; p < 0.02) (**Fig 10**). Healthcare professionals who perceived a lack of feedback were less likely to report errors than those who did not perceive it. Four studies reported that increased work experience is significantly associated with medical error reporting. The pooled AOR of medical error reporting for healthcare professionals with increasing work experience is 2.90 (95% CI: 1.25, 4.54; $I^2$ = 90.78%; p < 0.01) (**Fig 11**). Similarly, three articles reported a

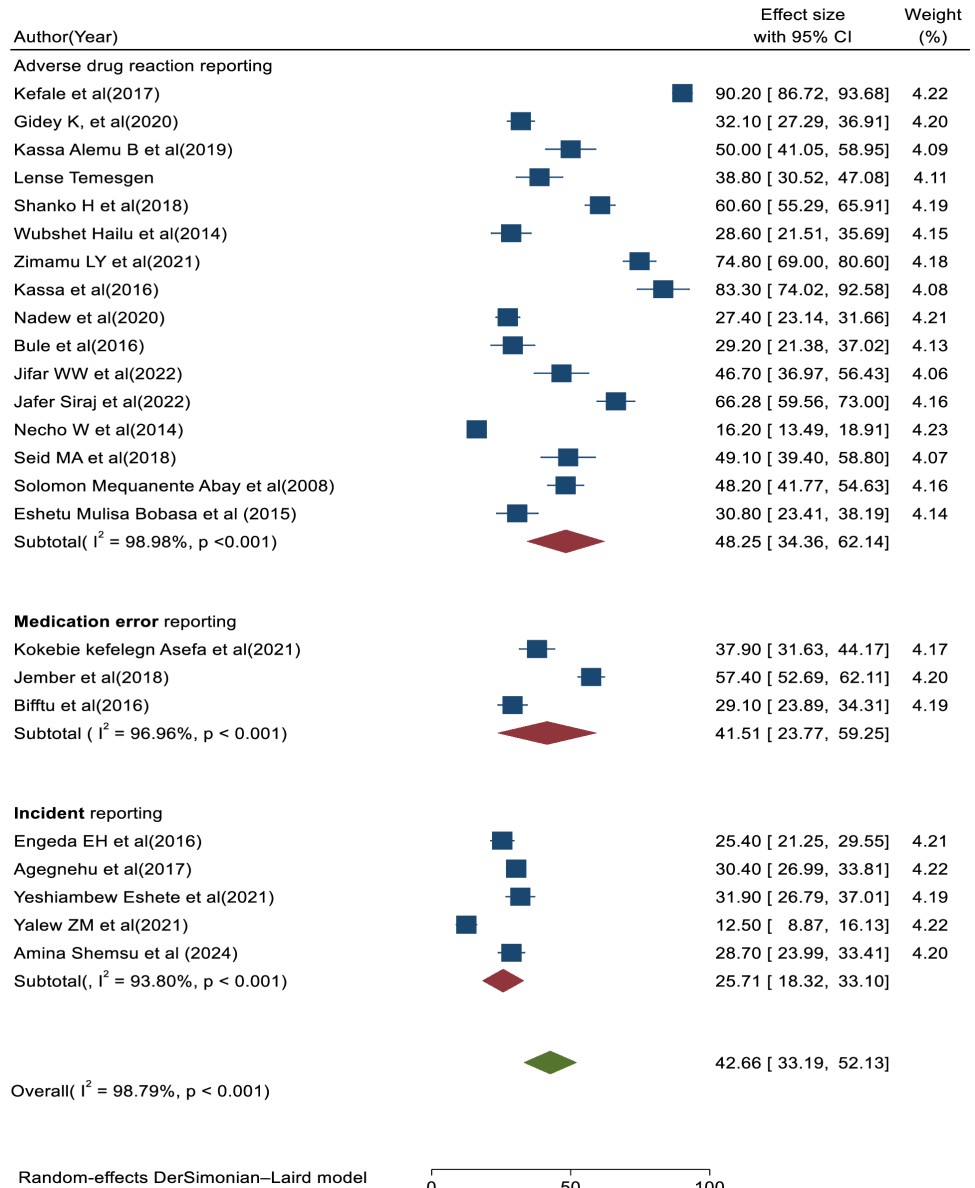

**Fig 7. Forest plot of prevalence of medical error reporting with 95%CIs of the subgroup analysis based on error reporting type, Ethiopia, 2024.**

significant association between healthcare professionals' gender and medical error reporting. The pooled AOR of medical error reporting for female professionals is 2.22 (95%CI: 0.10, 4.34; $I^2 = 97.09\%$; P = 0.00) (**Fig 12**). This suggests that female professionals were over two times more likely to report medical error than male professionals. Finally, three studies in the included articles showed that healthcare professionals' education status was significantly associated with medical error reporting. The pooled AOR for medical error reporting among healthcare professionals with higher education status is 3.20 (95% CI: 1.10, 5.30; $I^2 = 96.63\%$; P = 0.00) (**Fig 13**).

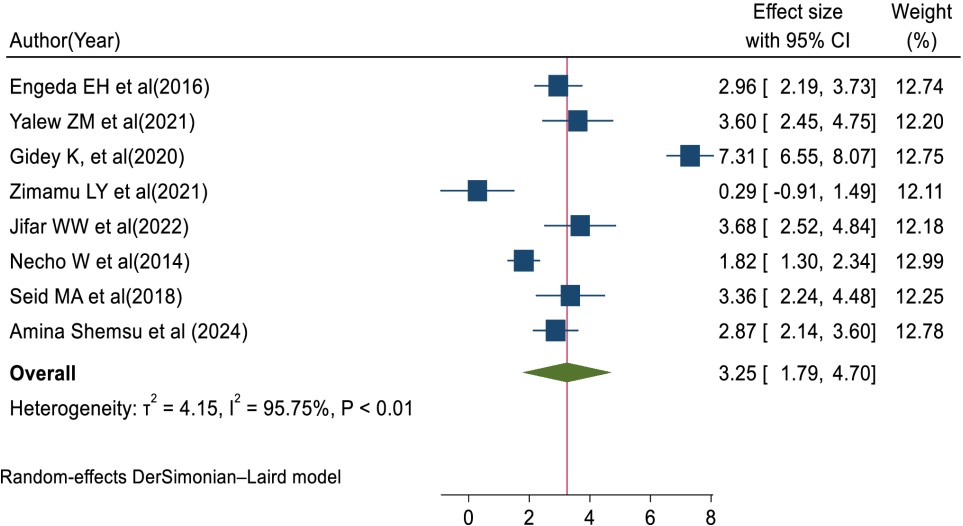

**Fig 8. Forest plot of adjusted odds ratio with 95%CIs of studies on the association of training and error reporting among health professional in Ethiopia, 2024.**

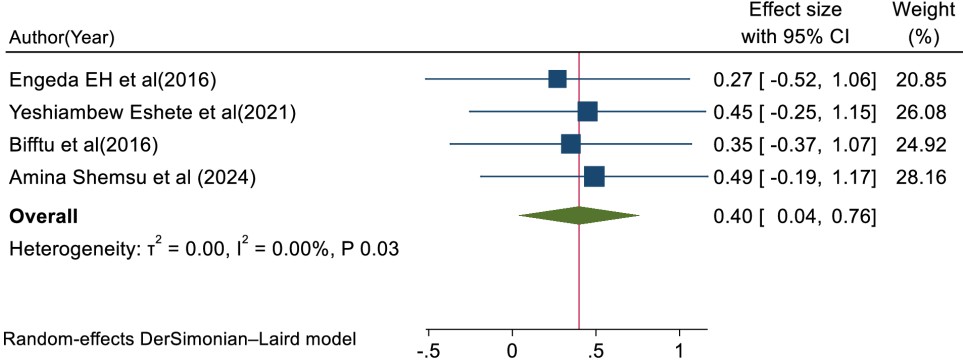

**Fig 9. Forest plot of adjusted odds ratio with 95%CIs of studies on the association of fear of administrative sanction and medical error reporting among health professional, 2024.**

## Discussion

This review aimed to assess the overall pooled prevalence of medical error reporting among healthcare professionals in Ethiopia, which was found to be 42.66% (95% CI: 33.19, 52.13; $I^2 = 98.79\%$, p<0.01). The result was lower than the findings from studies conducted in Kenya (58.1%) [62]), Nigeria (42.7%) [63], and Philippines (52%) [64]. The relatively low prevalence of error reporting in Ethiopia may be attributed to fear of punitive consequences, a lack of supportive systems, and inadequate feedback mechanisms [33]. Cultural and organizational factors may also play significant roles. Addressing these systemic challenges is crucial for fostering an environment that encourages error reporting and significantly improves patient safety [65]. However, without addressing these challenges, healthcare professionals may feel discouraged from reporting incidents, which can lead to repeated mistakes and unsafe practices, ultimately reducing patient safety and healthcare outcomes [66]. However, higher than study findings conducted in Iran [67,68], Malaysia [69], and South Africa [70]. This variation could be attributed to differences in how errors are defined and the focus on specific

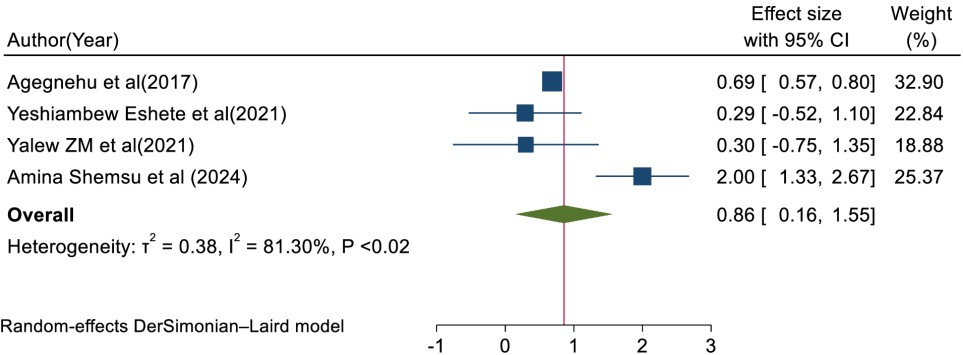

**Fig 10. Forest plot of adjusted odds ratio with 95%CIs of studies on the association of lack of feedback and medical error reporting among health professional in Ethiopia, 2024.**

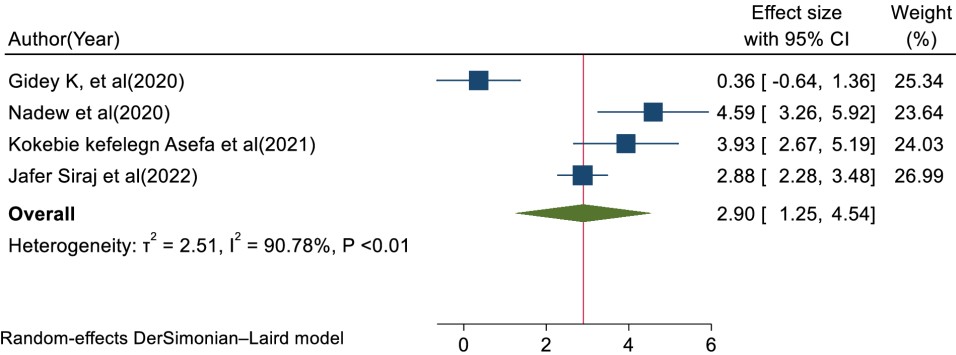

**Fig 11. Forest plot of adjusted odds ratio with 95%CIs of studies on the association of work experience and medical error reporting among health professional in Ethiopia, 2024.**

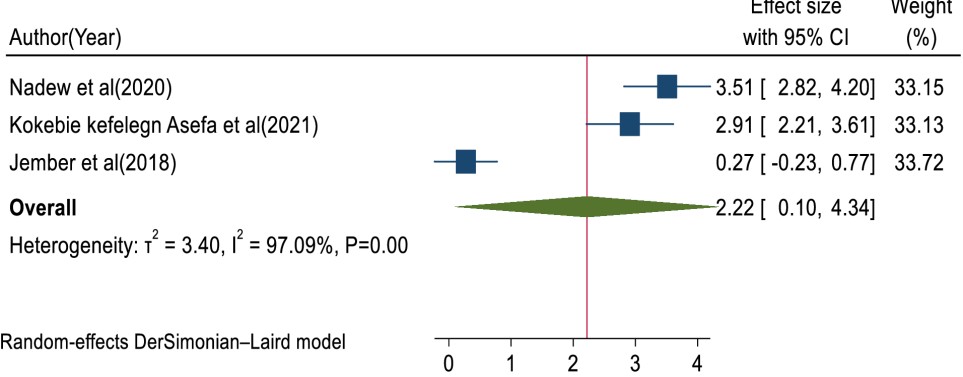

**Fig 12. Forest plot of the adjusted odds ratio with 95%CIs of studies on the association of professionals' gender and medical error reporting among health professionals in Ethiopia.**

types of error, which can significantly influence the prevalence of medical error reporting [71]. Moreover, in countries with poorly funded healthcare systems, there may be fewer resources allocated to quality control and error reporting mechanisms [72]. In addition to the aforementioned reasons, the variation might be attributed to differences in sample size, study

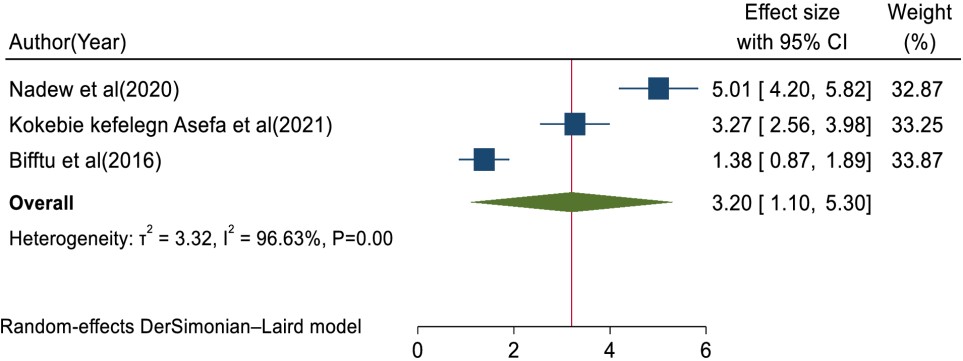

**Fig 13. Forest plot of the adjusted odds ratio with 95%CIs of studies on the association of educational status and medical error reporting among health professional in Ethiopia.**

duration, and settings [73]. For instance, the study in Iran was conducted seven years ago, while the study in South Africa was a primary research article.

In addition, the finding of this study revealed that health professionals who had received training on medical error reporting were more likely to report medical errors than those who did not receive training about error reporting. This study finding was in line with the study results conducted in Indonesia [74]. The likely reason for this association is that training on medical error reporting will increase health professionals' awareness about the error reporting systems and the importance of reporting and error identification [4]. In addition, it also improves the health professionals' skill in assessing the medical error to identify the causal factors and set methods to prevent their future occurrence of similar errors and be initiated to report new mistakes [33]. On top of that, trained professionals will recognize what medical error types are, how they report these mistakes, and where or for whom it will be reported [75].

Similarly, the results of this study showed that health professionals who feared administrative sanction were less likely to report medical errors compared to those who did not have such concerns. The study result is in line with findings of studies conducted in Canada [76], the USA [77], Iran [33], and Korea [78]. The likely reason for this association is that the punitive manner, as well as the inability to be protected from not being punished by their managers, makes professionals unwilling to report their mistakes, and they try to conceal the errors rather than disclosing them [26,79]. Furthermore, under the penal code of Ethiopia, causing harm to a patient due to medical error or negligence results in punishment, even if the action is unintentional. Healthcare professionals may hesitate to report medical error due to fear of the legal consequences [4], emphasizing the critical need for policy reforms that safeguard them from legal repercussions and foster a culture of more transparent and accurate error reporting. Like ways, in this study finding health professionals who perceived a lack of feedback were less likely to report medical error as compared with those who did not perceive it. The finding of this study is congruent with study findings conducted in USA [77], Iran [80], and Indonesia [74]. This could be attributed that the clinical risk management system (team) provides timely and structured feedback on the reported errors. This analyzed and detailed information should reveal the cause and strategies to prevent future occurrence of mistakes [81]. The significance of providing actionable response that improve system is recognized as a key factor to in encouraging future error reporting by healthcare professionals [82]. Similarly this study also reported that health professionals who had more work experience were more likely to report medical error than those who had less work experience. The result of this study also congruent with study finding conducted in Thailand [83]. The reason for this association could be health professionals with more experience would get a lesson from their daily duty in the healthcare settings which helps them to know more about medical error type, its definition and the

reporting system. Therefore, greater work experience leads to enhanced awareness and skills about the reporting system, which would results increased medical error reporting by professionals [27].

Regarding professionals' categories three articles were involved. Thus, the findings of these articles, rather than the pooled data, revealed that professionals such as nurses and pharmacists are more likely to report medical error than midwives, medical doctors and lab technician which was in line with the study result conducted in Thailand [83]. The reason of this association could be that nurses are closer to patient as compared with other health professionals and pharmacists are closer to medication as compared with the others. Thus, both nursing and pharmacy professionals are proximal to medical error and error reporting. Like ways in this study female health professionals were more likely to report medical error as compared to their counter-part (males). The finding was congruent with a study result conducted in Ireland [84] and Saudi Arabia [27]. This could be attributed that females tend to be less negligence than the male one in nature which may bring this association. Finally, the merged result of three primary studies reported that health professionals who had BSc and above were more likely to report medical error as compared to those who had diploma and below which was in line with a study result conducted in Saudi Arabia [27]. This might be linked to that advanced degree normally involves comprehensive training and education in patient safety and reporting protocols, which makes healthcare professionals more aware of the relevance of error reporting for patient safety culture [85]. On top of that, higher in education leads to enhanced critical thinking, and stronger sense of professional responsibility as well as accountability, motivating professionals to report medical error [86].

## Limitation and strength

Despite the high heterogeneity observed in the included studies, which may impact the interpretations of the findings, this is the first study to synthesize and analyze the results of multiple primary articles conducted in Ethiopia. The study provides stronger evidence on medical error reporting practices and its associated factors. Although all the studies were of high quality, it should be realized that they all employed a cross-sectional design and relied on self-reported data. Therefore, the study did not establish causal relationships between the outcome variable and the predictors. Additionally, the study could not perform subgroup analysis using the study year, professional categories, or reporting system type.

## Conclusions

The pooled prevalence of medical error reporting is suboptimal in Ethiopia, highlighting a significant gap in patient safety culture within healthcare institutions. Coordinated efforts are necessary to increase medical error reporting and improve the overall quality of patient care. Training, fear of administrative sanction, lack of feedback, work experience, professionals' gender, and educational status were identified as factors statistically significantly associated with medical error reporting. Addressing these issues by alleviating professionals' fear of consequences, ensuring legal protection for medical error reporting and establishing a robust reporting mechanism could significantly improve the reporting practices and ultimately enhance patient safety.

## Recommendation

**Policymakers**: Concerned bodies in the healthcare sector should prioritize the development of policies that foster supportive environment for medical error reporting, ensuring that healthcare professionals feel safe to report errors without fear of legal measures.

**Health institutions and stakeholders**: Healthcare Institutions should establish non-punitive error reporting system and provide clear guidelines on how reported errors will be handled. This would help mitigate fear of retribution and encourage more frequent reporting. Provide comprehensive training program focused on error definition, identification and the benefit of error reporting. Such training should emphasize the role of error reporting in enhancing patient safety and improve healthcare quality.

**Healthcare professionals** Professionals should be proactive in sharing insights, addressing concerns and implement strategies that enhance the overall patient safety culture in their institutions.

**Researchers**: Researchers are encouraged to assess the impact of various intervention on error reporting behavior. By focusing on strategies like anonymous reporting systems, studies could determine which approaches are most successful in enhancing reporting practice across healthcare settings. Additionally, the review highlights the lack of clarity regarding the source of heterogeneity among the included studies, so further research using rigorous study designs is needed to investigate its origin.

## Supporting information

**S1 File. PRISMA 2020 statement guidelines.**
(DOCX)

**S2 File. databases search string.**
(DOCX)

**S3 File. identified articles for this study.**
(DOCX)

**S4 File. extracted data for this study.**
(DOCX)

## Acknowledgments

We would like to give our deepest gratitude to Dr, Marelign Tilahun to his unrestricted support.

## Author contributions

**Conceptualization:** Yeshiambaw Eshetie, Tigabu Munye Aytenew, Bekalu Mekonen Belay, Mengistu Ewunetu, Yohannes Tesfahun, Demewoz Kefale.

**Data curation:** Yeshiambaw Eshetie, Yirgalem Abere, Tekalign Amera, Gebrie Kassaw Yirga, Melese Kebede.

**Formal analysis:** Yeshiambaw Eshetie, Tigabu Munye Aytenew, Mengistu Ewunetu, Demewoz Kefale.

**Funding acquisition:** Yeshiambaw Eshetie, Yirgalem Abere, Tekalign Amera, Yohannes Tesfahun, Melese Kebede.

**Investigation:** Yeshiambaw Eshetie, Yirgalem Abere, Mengistu Ewunetu, Gebrie Kassaw Yirga, Demewoz Kefale.

**Methodology:** Yeshiambaw Eshetie, Bekalu Mekonen Belay, Tekalign Amera.

**Project administration:** Yeshiambaw Eshetie, Mengistu Ewunetu, Yohannes Tesfahun, Melese Kebede.

**Resources:** Yeshiambaw Eshetie, Tekalign Amera, Gebrie Kassaw Yirga, Melese Kebede, Demewoz Kefale.

**Software:** Yeshiambaw Eshetie, Bekalu Mekonen Belay, Mengistu Ewunetu.

**Supervision:** Yeshiambaw Eshetie, Tigabu Munye Aytenew, Tekalign Amera, Yohannes Tesfahun, Demewoz Kefale.

**Validation:** Yeshiambaw Eshetie, Yirgalem Abere, Bekalu Mekonen Belay, Mengistu Ewunetu, Gebrie Kassaw Yirga.

**Visualization:** Yeshiambaw Eshetie, Tigabu Munye Aytenew, Yirgalem Abere, Tekalign Amera, Melese Kebede.

**Writing – original draft:** Yeshiambaw Eshetie, Tigabu Munye Aytenew, Bekalu Mekonen Belay, Mengistu Ewunetu, Yohannes Tesfahun.

**Writing – review & editing:** Yeshiambaw Eshetie, Tigabu Munye Aytenew, Yirgalem Abere, Tekalign Amera, Gebrie Kassaw Yirga, Melese Kebede, Demewoz Kefale.

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
