## [Decision Letter · Decision Letter 0]

11 Oct 2024

PONE-D-24-27747Magnitude of medical error reporting and associated factors among health professional in Ethiopia. Systematic Review and Meta-AnalysisPLOS ONE

Dear Dr. Eshetie,

Thank you for submitting your manuscript to PLOS ONE. After careful consideration, we feel that it has merit but does not fully meet PLOS ONE’s publication criteria as it currently stands. Therefore, we invite you to submit a revised version of the manuscript that addresses the points raised during the review process.

We look forward to receiving your revised manuscript.

Kind regards,

Zelalem Belayneh Muluneh

Academic Editor

PLOS ONE

**Journal Requirements:**

5. Please remove your figures from within your manuscript file, leaving only the individual TIFF/EPS image files, uploaded separately. These will be automatically included in the reviewers’ PDF.

6. As required by our policy on Data Availability, please ensure your manuscript or supplementary information includes the following: 

Reviewers' comments:

Reviewer's Responses to Questions

**Comments to the Author**

1. Is the manuscript technically sound, and do the data support the conclusions?

Reviewer #1: Partly

Reviewer #2: Partly

2. Has the statistical analysis been performed appropriately and rigorously? 

Reviewer #1: Yes

Reviewer #2: Yes

3. Have the authors made all data underlying the findings in their manuscript fully available?

Reviewer #1: Yes

Reviewer #2: No

4. Is the manuscript presented in an intelligible fashion and written in standard English?

Reviewer #1: No

Reviewer #2: No

5. Review Comments to the Author

**Reviewer #1:**  See the attached additional note

This systematic review and meta-analysis investigate the crucial issue of medical error reporting among health professionals in Ethiopia, examining both prevalence and associated factors.

The rationale for combining studies on different types of medical error reporting (medication errors, adverse drug reactions, incidents) requires further clarification. Specifically, the authors should clearly explain the shared characteristics and distinctions between these error types. This will help readers understand why these categories were grouped for analysis. The authors also need to provide a strong justification for analysing these diverse error types together. For instance, are there common underlying factors influencing reporting rates across these categories? Does combining them offer a more comprehensive understanding of error-reporting behaviour in Ethiopia?

The authors need to rephrase the research question to be more focused for example "What is the prevalence of medical error reporting among health professionals in Ethiopia, and what are the associated factors?". The author should consider providing specific, objective criteria for excluding studies based on their results. What is the author's rationale for using the JBI Critical Appraisal Checklist for prevalence studies? There is a need to clarify the approach to analysing AORs in the meta-analysis, considering potential issues related to combining adjusted effect estimates from different studies. There is no table summarizing the characteristics of included studies and reported heterogeneity measures for each pooled analysis.

Finally, ensure the manuscript is free of grammatical errors and inconsistencies. The manuscript requires major revisions before it can be considered for publication. Addressing the major flaws, particularly those related to the analysis and reporting of AORs and the clarity of the results, is essential. Attending to the minor flaws would further strengthen the manuscript and improve its overall quality

**Reviewer #2: ** The manuscript is a systematic review and meta-analysis aimed at determining the prevalence of medical error reporting and identifying associated factors among health professionals in Ethiopia. By synthesizing data from 24 eligible studies and offering insights into the barriers and facilitators of medical error reporting, it provides an important contribution to the existing literature.

1. Clarity and Language:

o The manuscript suffers from frequent grammatical errors, awkward phrasing, and incorrect word usage (e.g., “professional sex” instead of "gender"). These issues detract from the readability and professionalism of the paper.

o Some sentences are confusing, and the logic is difficult to follow in parts, particularly in the introduction and discussion sections. For example, the line “Medical error is a failure of a ‘planned action to be completed as intended or the use of a wrong plan to achieve aim’” is not clearly articulated.

Recommendation: The manuscript should undergo thorough language editing for clarity, grammar, and word choice.

2. Methodological Details:

o Data Extraction: While the authors mention using a data extraction form, they provide limited details about how reviewer disagreements were handled. A more thorough explanation of the steps taken to resolve conflicts would enhance the robustness of the review process.

o Risk of Bias: The authors mention using the JBI Critical Appraisal Checklist but do not detail how the included studies were assessed. For example, it is unclear if the analysis weighted the low-risk, moderate-risk, and high-risk studies differently.

Recommendation: Include a table with the quality assessments of all included studies and explain how potential biases were accounted for in the meta-analysis.

3. Discussion and Interpretation:

o The discussion would benefit from more profound reflection on the implications of the findings, particularly in the global context. Comparisons are made to Kenya, Nigeria, and other countries, but the reasons for the differences are not sufficiently explored. Additionally, the implications of the findings for health policy in Ethiopia are not discussed in depth.

o Causality: While several factors (e.g., training, fear of administrative sanctions) are identified as being associated with reporting behaviour, the authors should clarify that causality cannot be inferred from the cross-sectional studies included in the meta-analysis.

Recommendation: Expand the discussion to address the implications of the findings more comprehensively, considering health policy and the practical applications of the results.

4. Heterogeneity and Subgroup Analyses:

o The heterogeneity (I2 = 98.79%) is very high, indicating significant variability among the studies. Although the authors conduct subgroup analyses, the explanation of the sources of heterogeneity could be more detailed. For instance, the potential impact of different healthcare settings, reporting systems, or healthcare worker roles on heterogeneity is not fully explored.

Recommendation: Provide a more in-depth discussion of the possible causes of heterogeneity and whether future research should target these areas to reduce variability.

5. Figures and Tables:

o While the forest plots and subgroup analyses are helpful, the presentation of the data could be more organised. For instance, Table 1, which presents the characteristics of included studies, is somewhat cluttered and challenging to read.

Recommendation: Consider reformatting the tables to improve clarity and readability.

Conclusion:

The manuscript addresses an important topic and contributes to understanding medical error reporting in Ethiopia. However, several areas need improvement, particularly in the clarity of writing, methodological transparency, and depth of analysis. With revisions, the paper could provide even greater insight into the factors influencing medical error reporting and their implications for healthcare policy.

6. PLOS authors have the option to publish the peer review history of their article (what does this mean? ). If published, this will include your full peer review and any attached files.

**Do you want your identity to be public for this peer review?** For information about this choice, including consent withdrawal, please see our Privacy Policy .

Reviewer #1: No

Reviewer #2: No

---

## [Author Response · Author response to Decision Letter 1]

15 Nov 2024

Point by point response to editor and/or reviewer(s)

1. Appropriateness of Study Design

“The rationale for including studies on different types of medical error reporting (medication errors, adverse drug reactions, incidents) could be more robust. The authors should elaborate on the similarities and differences between these types and justify their combined analysis.”

Authors’ response: We thank you for your insightful input and concern regarding the inclusion of various types of medical error reporting and the rationale of the combined analysis. We acknowledge your comment on the need for robust explanation on the similarities and differences among medication error reporting, adverse drug reaction reporting and incident reporting (see page 3 and 4). To name few:-

1. Diversity of error reports: All adverse drug reaction reporting, medication error reporting and incident reporting allows healthcare managers to capture a comprehensive view of patient safety issues at the hospital level. Each type can reveals different dimensions of healthcare challenges in each working unit and department of healthcare providers.

2. Interconnectedness: These types of errors are often interconnected. For instance, a medication error may lead to an adverse drug reaction and both can be categorized as an incident affecting patient safety culture that leads reduction in patient outcome of the healthcare institutions. Therefore, analyzing them together can identify patterns and systemic vulnerabilities of the healthcare delivering institution rather than focusing each fragmented types of error reporting.

3. Regarding report responsibility: Doctors, nurses, and midwives are often reporting medication error and adverse drug reactions as part of their clinical responsibilities and pharmacists also play a crucial role in identifying and reporting medication errors as well as adverse drug reactions. Thus, there is role overlapping rather than distinction among healthcare professionals that reveals the need of using a comprehensive error reporting system.

4. Justification of the combined analysis: This combined analysis emphasize the benefit of holistic approach to common root cause and developing effective strategies for prevention that can be applied across these different type of errors, and used to bring quality cares in the organization as whole. This combined analysis employed for regulation and accreditation standard preparations by considering healthcare institution as one entity rather than various departments separately.

5. Implication: Understanding these errors collectively can reveal need of police change in this sector aimed at improving healthcare services provided and reducing unwanted events.

Revision in the manuscript: We incorporated these clarifications and justifications in the revised manuscript to offer more robust rationale for the combined analysis.

2. Clarity and Relevance of Research Question

“The introduction mentions different types of medical errors such as "medication errors, adverse reactions, and incidents" but doesn't clarify whether the review will focus on all of them or a specific type. The primary research question, while generally clear, could be more concisely” stated. Defining the scope of "medical error reporting" more precisely in the introduction would enhance clarity.

Authors’ response: we would like say thank you for your constructive comment on the clarity and relevance of the question of this study. We modified the introduction to clearly indicate that the review encompasses all types of medical errors mentioned earlier and we revised it accordingly. We acknowledge your suggestion to make the research question more concise. So, based on your comment we revised it to enhance clarity as well as ensure it concisely captures the essence of the review. Regarding the need of defining the scope of medical error reporting we agree that a clearer explanation improves the reader’s understanding and enhance overall clarity of the review. We have incorporated your valuable comments in our revised manuscript (see page 3, line-16 and 17).

---

## [Decision Letter · Decision Letter 1]

6 Jan 2025

PONE-D-24-27747R1Prevalence and its associated factors of medical error reporting among health professionals in Ethiopia: Systematic Review and Meta-analysis.PLOS ONE

Dear Dr. Eshetie,

Thank you for submitting your manuscript to PLOS ONE. After careful consideration, we feel that it has merit but does not fully meet PLOS ONE’s publication criteria as it currently stands. Therefore, we invite you to submit a revised version of the manuscript that addresses the points raised during the review process.

We look forward to receiving your revised manuscript.

Kind regards,

Zelalem Belayneh Muluneh

Academic Editor

PLOS ONE

Journal Requirements:

Additional Editor Comments (if provided):

Reviewers' comments:

Reviewer's Responses to Questions

**Comments to the Author**

Reviewer #2: All comments have been addressed

Reviewer #3: (No Response)

2. Is the manuscript technically sound, and do the data support the conclusions?

Reviewer #2: Partly

Reviewer #3: Yes

3. Has the statistical analysis been performed appropriately and rigorously? 

Reviewer #2: No

Reviewer #3: Yes

4. Have the authors made all data underlying the findings in their manuscript fully available?

Reviewer #2: Yes

Reviewer #3: Yes

5. Is the manuscript presented in an intelligible fashion and written in standard English?

Reviewer #2: No

Reviewer #3: No

6. Review Comments to the Author

Reviewer #2: Second review of “Prevalence and its associated factors of medical error reporting among health professionals in Ethiopia: Systematic Review and Meta-analysis.”

The authors have provided comprehensive and thoughtful responses to the reviewers' comments, demonstrating a commitment to improving the manuscript. They have addressed the key concerns related to the definition and scope of medical error reporting, methodological clarity, data analysis, and presentation of results. The revisions outlined in their responses should enhance the manuscript's clarity, rigour, and impact. In the following, I evaluate their responses and how they addressed the issues raised in our first review. I follow that with a second round of review, which I believe is necessary for the manuscript to be publishable.

Detailed Evaluation:

1. Definition and Scope of Medical Error Reporting:

o Authors' Response:

Elaborated on the similarities and differences between medication error reporting, adverse drug reaction reporting, and incident reporting.

Justified the combined analysis by discussing interconnectedness, overlapping responsibilities among healthcare professionals, and the benefits of a holistic approach.

Incorporated these explanations into the revised manuscript (pages 3 and 4).

o Evaluation:

The authors have effectively addressed the need for a more explicit definition and scope of medical error reporting.

Providing detailed justifications and incorporating them into the manuscript has enhanced the reader's understanding of the rationale behind the combined analysis.

2. Rephrasing the Research Question:

o Authors' Response:

Modified the research questions and the title to be more focused and concise, as suggested.

o Evaluation:

The revised research question now clearly articulates the objectives of the study.

A focused research question enhances the coherence and direction of the manuscript.

3. Exclusion Criteria and Study Selection:

o Authors' Response:

Revised the exclusion criteria to offer specific and objective parameters, enhancing clarity and reducing potential bias (page 6, lines 1-2).

o Evaluation:

Providing explicit inclusion and exclusion criteria strengthens the methodological rigour of the review.

4. Use of the JBI Critical Appraisal Checklist:

o Authors' Response:

Justified using the Joanna Briggs Institute (JBI) Critical Appraisal Checklist.

Included a detailed explanation of how studies were assessed and assigned risk of bias levels (pages 7 and 8).

Added appropriate citations.

o Evaluation:

The authors have addressed the concern by clarifying their methodology and referencing the appropriate tools.

Recommendation: Present the quality assessment results in a table summarising each study's score on the checklist items. This will enhance transparency and allow readers to assess the quality of the included studies.

5. Approach to Analyzing Adjusted Odds Ratios (AORs):

o Authors' Response:

Acknowledged potential issues with combining AORs from studies with different adjustments.

Explained the use of a random-effects model to account for variability.

Conducted sensitivity analyses and robust quality assessments.

o Evaluation:

The authors have taken appropriate steps to address variability in AORs.

Recommendation:

Clearly describe how the differences in confounders adjusted for in each study were handled in the methods section.

Discuss the limitations of combining AORs with varying adjustments in the discussion section.

6. Presentation of Study Characteristics and Heterogeneity Measures:

o Authors' Response:

Included comprehensive summaries of the included studies' characteristics.

Added heterogeneity measures for each pooled value in the subgroup analyses (page 9, lines 9-19).

o Evaluation:

This addition improves the manuscript by providing essential context and facilitating study comparisons.

7. Clarity and Language:

o Authors' Response:

Acknowledged the presence of grammatical errors and inconsistencies.

Addressed these issues in the revised manuscript.

8. Data Extraction and Handling Reviewer Disagreements:

o Authors' Response:

Included explanations of how disagreements during data extraction were resolved (page 7, lines 1-5).

o Evaluation:

This addition enhances the transparency and robustness of the review process.

9. Discussion and Interpretation:

o Authors' Response:

Expanded the discussion to explore potential reasons for differences in prevalence between Ethiopia and other countries.

Considered factors such as healthcare system structure, reporting culture, and types of errors addressed (pages 12-14).

Corrected redundancy in the title and throughout the manuscript.

o Evaluation:

The enriched discussion provides valuable insights into the context and implications of the findings.

10. Heterogeneity and Subgroup Analyses:

o Authors' Response:

Addressed the high heterogeneity observed in the study.

Provided more in-depth discussion on possible causes of heterogeneity (pages 10 and 11).

o Evaluation:

Acknowledging and exploring heterogeneity is essential in meta-analysis.

11. Figures and Tables:

o Authors' Response:

Improved the presentation of data in tables and figures for better clarity and readability.

o Evaluation:

Well-designed tables and figures enhance the reader's understanding and interpretation of the data.

The response by the authors addressed crucial deficiencies in the previous version of the manuscript. However, there are still areas that need further improvement. I tried my best to create detailed guidelines for the authors to follow:

1. Abstract

Issues:

• Clarity and Structure: The abstract still contains grammatical errors and could be more concise.

• Incomplete Information: The abstract mentions that 24 studies were included but does not specify the types of medical errors or the professions of health workers.

Recommendations:

• Revise for Clarity: Proofread the abstract to correct grammatical errors and improve sentence structure.

• Include Specifics: Provide brief information on the types of medical errors studied and the health professions involved.

• Structure: Ensure the abstract follows a clear structure: Background, Methods, Results, and Conclusion.

Example Revision:

Background: Medical errors pose a significant threat to patient safety worldwide. Reporting these errors is crucial for reducing healthcare-related mistakes. However, the prevalence of medical error reporting and its associated factors are not well established in Ethiopia.

Methods: We conducted a systematic review and meta-analysis of cross-sectional studies assessing the prevalence and factors associated with medical error reporting among health professionals in Ethiopia. We also performed an extensive literature search using databases such as PubMed, Google Scholar, and Web of Science. The pooled prevalence was calculated using a random-effects model.

Results: Twenty-four studies involving 6,745 health professionals were included. The overall pooled prevalence of medical error reporting was 42.66% (95% CI: 33.19%, 52.13%). Factors significantly associated with medical error reporting included receiving training (AOR = 3.25), fear of administrative sanctions (AOR = 0.40), lack of feedback (AOR = 0.86), work experience (AOR = 2.90), gender (AOR = 2.22), and education level (AOR = 3.20).

Conclusion: Medical error reporting among health professionals in Ethiopia is relatively low. Targeted interventions, such as training programs and creating non-punitive reporting environments, are needed to improve reporting practices.

2. Introduction

Issues:

• Grammar and Clarity: The introduction contains grammatical errors which affect readability.

• Flow and Coherence: The narrative jumps between ideas without smooth transitions, making it difficult to follow.

• Literature Review: While international studies are cited, there is limited critical analysis of existing literature specific to Ethiopia, and the research gap is unclear.

Recommendations:

• Proofread for Language Clarity: Correct grammatical errors and improve sentence structure for better readability.

• Enhance Flow: Use transitional phrases to connect ideas and ensure a logical progression of thought.

• Expand Literature Review:

o Critical Analysis: Provide a critical synthesis of existing studies on medical error reporting in Ethiopia and globally.

o Identify Gaps: Clearly articulate the gaps in the literature that your study aims to fill.

o Justify the Study: Emphasize the importance of a systematic review and meta-analysis to consolidate fragmented findings and provide national-level insights.

Example of Improved Paragraph:

Original: "Medical errors (ME) are mistakes that doctors, nurses, and other health professionals commit during the process of diagnosing, treating, caring and monitoring patients [1, 2]. In general, it refers to the failure of healthcare professional to execute a planned action as intended, or the application of an inappropriate plan to achieve a desired outcome [3]."

Revised: "Medical errors (MEs) refer to mistakes made by healthcare professionals—including doctors, nurses, and others—during the processes of diagnosis, treatment, care, and patient monitoring [1,2]. They encompass failures to execute planned actions as intended or the use of incorrect plans to achieve desired outcomes [3]. These errors pose significant threats to patient safety globally."

3. Methods

Issues:

• Search Strategy: The description lacks detail about search terms, specific inclusion/exclusion criteria, and the search timeframe.

• Data Extraction and Quality Assessment: The process is briefly described but could benefit from more detail, including how reviewer disagreements were resolved.

• Risk of Bias Assessment: There is limited discussion on how the quality of the included studies was assessed using the JBI checklist.

Recommendations:

• Provide Detailed Search Strategy: Include the specific search terms used, the time frame of the search, and any language restrictions.

• Inclusion/Exclusion Criteria: Specify the criteria used to include or exclude studies, such as study design, population, outcomes measured, and settings.

• Data Extraction Process: Elaborate on the data extraction process, including how data were managed and stored and how discrepancies were handled.

• Quality Assessment: Provide more detail on how the JBI checklist was applied, including the criteria assessed and how the quality scores influenced the inclusion of studies.

• Subgroup Analysis: In the methods and results, provide more detail on the subgroup analyses conducted and discuss their implications.

• Statistical Methods: Provide more detailed descriptions of the statistical methods used, including how the random-effects model was applied and how heterogeneity was quantified and addressed.

Example Addition:

"In our literature search, we used a combination of MeSH terms and free-text keywords such as 'medical error,' 'error reporting,' 'health professionals,' and 'Ethiopia.' Without language restrictions, searches were conducted from April 10 to June 10, 2024. Two independent reviewers screened titles and abstracts for relevance, and any disagreements were resolved through discussion with a third reviewer."

4. Results

Issues:

• Presentation of Findings: The results section could benefit from more explicit organisation and detailed descriptions of the included studies.

• Heterogeneity and Publication Bias: While heterogeneity is acknowledged, the implications are not fully discussed. Methods used to address heterogeneity (e.g., subgroup analysis) need more explanation.

• Associated Factors: The analysis of associated factors is presented but lacks depth in interpretation.

Recommendations:

• Organise Results with Subheadings: Use subheadings such as 'Study Selection,' 'Characteristics of Included Studies,' 'Pooled Prevalence,' 'Heterogeneity Assessment,' 'Publication Bias,' 'Sensitivity Analysis,' 'Subgroup Analysis,' and 'Factors Associated with Medical Error Reporting.'

• Discuss Heterogeneity: Provide a more detailed discussion on the high heterogeneity observed, potential reasons, and how it was addressed in the analysis.

• Interpret Associated Factors: Offer in-depth interpretation of the findings related to associated factors, including potential mechanisms and implications.

Example of Improved Presentation:

Under 'Factors Associated with Medical Error Reporting':

"Training was significantly associated with increased medical error reporting (AOR = 3.25, 95% CI: 1.79, 4.70). This suggests that health professionals who received training were over three times more likely to report errors than those who did not. This finding highlights the importance of educational interventions in promoting error reporting."

5. Discussion

Issues:

• Interpretation of Results: The discussion provides comparisons with other countries but lacks depth in exploring the reasons behind the differences.

• Link to Existing Literature: The study findings are limitedly integrated with existing literature, particularly in explaining why certain factors are associated with medical error reporting.

• Limitations: The limitations section is brief and does not fully consider the impact of the study design and heterogeneity.

Recommendations:

• Deepen Interpretation: Explore potential reasons for Ethiopia's low prevalence of error reporting, considering cultural attitudes, systemic barriers, and policy implications.

• Integrate with Literature: Discuss how your findings align or contrast with previous studies within Ethiopia and internationally and what this means for understanding medical error reporting.

• Expand on Limitations: Provide a thorough discussion of the study's limitations, including the high heterogeneity, potential publication bias, reliance on self-reported data, and the limitations inherent in cross-sectional studies.

Example Addition:

"The relatively low prevalence of medical error reporting in Ethiopia compared to countries like Kenya and Nigeria may be attributed to fear of punitive consequences, lack of supportive reporting systems, and inadequate feedback mechanisms [66,67]. Cultural factors and organisational barriers may also play significant roles [Reference]. Addressing these systemic issues is crucial for improving reporting rates and patient safety."

6. Conclusion

Issues:

• Clarity and Impact: The conclusion restates the findings but could be more impactful by emphasising practical implications and specific recommendations.

Recommendations:

• Emphasise Practical Implications: Highlight how the findings can inform policy and practice, suggesting specific interventions tailored to the Ethiopian context.

• Call to Action: Encourage stakeholders to implement strategies to improve medical error reporting, such as policy changes, training programs, and developing non-punitive reporting cultures.

Example Revision:

"In conclusion, medical error reporting among health professionals in Ethiopia is suboptimal, indicating a critical need for interventions. Implementing comprehensive training programs, establishing non-punitive reporting environments, and improving feedback mechanisms are essential to enhance reporting practices. These measures can contribute significantly to patient safety and healthcare quality in Ethiopia."

7. Language and Grammar

Issue:

• The manuscript contains grammatical errors, typos, and awkward phrases that affect readability.

Recommendation:

• Professional Editing: Consider engaging a professional editor or utilising language editing services to improve clarity, grammar, and overall readability.

8. Formatting and References

Issues:

• Reference Style: The references are inconsistently formatted and may not adhere to the Vancouver style applied throughout the manuscript.

• Citations: Some in-text citations lack corresponding references, and vice versa.

Recommendations:

• Consistent Formatting: Ensure all references are formatted consistently according to the journal's guidelines, including correctly using italics, punctuation, and abbreviations.

• Complete Citations: Cross-check all in-text citations with the reference list to ensure completeness and accuracy.

• Use Reference Management Software: Utilise software like EndNote or Zotero to manage references and maintain consistency.

Examples:

1. Inconsistent Formatting of References:

o Example 1:

Original Reference:

3. Grober, E.D. and J.M. Bohnen, Defining medical error. Can J Surg, 2005. 48(1): p. 39-44.

Corrected Reference (assuming Vancouver style):

3. Grober ED, Bohnen JM. Defining medical error. Can J Surg. 2005;48(1):39-44.

o Example 2:

Original Reference:

17. Organization, W.H., Patient safety incident reporting and learning systems: technical report and guidance. 2020.

Corrected Reference:

17. World Health Organization. Patient safety incident reporting and learning systems: technical report and guidance [Internet]. Geneva: World Health Organization; 2020 [cited YEAR MONTH DAY]. Available from: URL or DOI if applicable.

o Example 3:

Original Reference:

44. Page, M.J., et al., The PRISMA 2020 statement: an updated guideline for reporting systematic reviews. BMJ, 2021. 372.

Corrected Reference:

44. Page MJ, McKenzie JE, Bossuyt PM, Boutron I, Hoffmann TC, Mulrow CD, et al. The PRISMA 2020 statement: an updated guideline for reporting systematic reviews. BMJ. 2021;372:n71.

2. Missing In-text Citations:

o Example:

"For instance, it occurs at a rate of 50.5%-95.5% in Iran, and the participants reported that they experienced at least one medical error in a year." There is no in-text citation provided to support this statement despite mentioning specific data that requires referencing.

3. Incomplete References:

o Example:

65. Bati Tariku, E.M., Shibiru Tesema, HEALTH PROFESSIONALS’ KNOWLEDGE, ATTITUDE AND PRACTICES TOWARDS ADVERSE DRUG REACTION REPORTING IN NEKEMTE HOSPITAL, ETHIOPIA. Medical review, 2015. 7(1): p. 041-046.

Corrected Reference:

65. Tariku B, Mulisa E, Tesema S. Health professionals’ knowledge, attitude, and practices towards adverse drug reaction reporting in Nekemte Hospital, Ethiopia. Med Rev. 2015;7(1):41-46.

Reviewer #3: I would like to commend the authors for their efforts in contributing to evidence-based practice by providing the first pooled evidence for Ethiopia. However, several concerns need to be addressed:

① Language Editing: The manuscript requires extensive language editing to improve clarity, grammar, and readability. Consider seeking professional language editing services or support.

② Abstract: The abstract should be more concise.

1. The introduction should provide the background in 1-2 sentences and clearly state the study's objective. Avoid repeating the objective in both the introduction and methods sections. The authors should focus on using the word limit of the abstract effectively.

2. In the Methods section, specify the duration of the database search and the types of studies included.

3. In the Results section, mention the total number of records identified through the database and manual search.

4. The results are not precise. For example,

- Clarify what is meant by "training"—does it refer to a lack of training on medical error reporting?

- Specify the context of "work experience" and "education" to avoid ambiguity.

- Clearly state which sex is being referred.

③ Introduction:

1. The introduction needs to be more concise and well-structured. Ensure the flow of information is logical, and avoid repeating the same content across paragraphs (notably in paragraphs 1-3).

2. In Paragraph 1, provide numeric evidence (preferably global data) to substantiate the statement: “Even though the majority of errors do not lead to an apparent adverse effect, a considerable number of patients either suffer permanent harm or die every year worldwide as a result of these mistakes.”

3. The statement (paragraph 4)“Despite the numerous advantages of error reporting, the magnitude of underreporting remains high in different countries around the world” is supported with data from only two countries. Including global-level data or examples from diverse regions would enhance the credibility and comprehensiveness of this claim.

④ Methods:

1. Provide details of the search strategy used for at least one database to improve transparency and replicability.

2. To enhance clarity and a structured overview of the inclusion and exclusion criteria, include a table summarizing the PICOS criteria (Population, Intervention, Control, Outcome, and Study design).

3. The study aims to determine the pooled prevalence of medical error reporting and identify associated factors among health professionals in Ethiopia. The authors should justify why studies published in the national languages of Ethiopia were not included.

⑤ Results:

1. The authors use the terms ‘sex’ and ‘gender’ interchangeably. These terms should be unified throughout the manuscript to avoid confusion.

2. Even the results section does not clearly explain how sex is associated with medical error reporting (was the prevalence higher among men or women?)

3. The term “education level” is unclear. If the authors are referring to the professional category (e.g., nurses, physicians, etc.), it would be more appropriate to use this terminology rather than dividing participants into educational categories.

⑥ Discussion:

1. Include a section at the end of the discussion offering recommendations for future research and practical implications. Authors could add a discussion on the current legislation and policies related to medical error reporting in Ethiopia, highlight any loopholes or gaps in the system, and propose methods to improve error reporting practices.

7. PLOS authors have the option to publish the peer review history of their article (what does this mean? ). If published, this will include your full peer review and any attached files.

**Do you want your identity to be public for this peer review?** For information about this choice, including consent withdrawal, please see our Privacy Policy .

Reviewer #2: No

Reviewer #3: No

---

## [Author Response · Author response to Decision Letter 2]

29 Jan 2025

Point by point response for editor

Journal Name PLOS ONE

Manuscript number: D-24-27747

Title: Prevalence and its associated factors of medical error reporting among healthcare professionals in Ethiopia: Systematic Review and Meta-Analysis.

Thank you, editor, for the opportunity to submit the revised version of the manuscript and to correct edition problems and respond questions.

---

## [Decision Letter · Decision Letter 2]

8 May 2025

Prevalence and its associated factors of medical error reporting among healthcare professionals in Ethiopia: Systematic Review and Meta-analysis.

PONE-D-24-27747R2

Dear Dr. Eshetie,

We’re pleased to inform you that your manuscript has been judged scientifically suitable for publication and will be formally accepted for publication once it meets all outstanding technical requirements.

Kind regards,

Zelalem Belayneh

Academic Editor

PLOS ONE

Additional Editor Comments (optional):

Reviewers' comments:

Reviewer's Responses to Questions

**Comments to the Author**

1. If the authors have adequately addressed your comments raised in a previous round of review and you feel that this manuscript is now acceptable for publication, you may indicate that here to bypass the “Comments to the Author” section, enter your conflict of interest statement in the “Confidential to Editor” section, and submit your "Accept" recommendation.

Reviewer #2: All comments have been addressed

2. Is the manuscript technically sound, and do the data support the conclusions?

Reviewer #2: Yes

3. Has the statistical analysis been performed appropriately and rigorously? 

Reviewer #2: Yes

4. Have the authors made all data underlying the findings in their manuscript fully available?

Reviewer #2: Yes

5. Is the manuscript presented in an intelligible fashion and written in standard English?

Reviewer #2: Yes

6. Review Comments to the Author

Reviewer #2: Dear Authors,

Congratulations on your manuscript. The current version of your manuscript addresses the issues raised by the reviewers. The manuscript looks like a proper SR and meta-analysis.

7. PLOS authors have the option to publish the peer review history of their article (what does this mean? ). If published, this will include your full peer review and any attached files.

**Do you want your identity to be public for this peer review?** For information about this choice, including consent withdrawal, please see our Privacy Policy .

Reviewer #2: No

---

## [Editor Report · Acceptance letter]

PONE-D-24-27747R2

PLOS ONE

Dear Dr. Eshetie,

I'm pleased to inform you that your manuscript has been deemed suitable for publication in PLOS ONE. Congratulations! Your manuscript is now being handed over to our production team.

Kind regards,

on behalf of

Mr. Zelalem Belayneh

Academic Editor

PLOS ONE